# A Pilot Study of Vaccine Therapy with Multiple Glioma Oncoantigen/Glioma Angiogenesis-Associated Antigen Peptides for Patients with Recurrent/Progressive High-Grade Glioma

**DOI:** 10.3390/jcm8020263

**Published:** 2019-02-20

**Authors:** Ryogo Kikuchi, Ryo Ueda, Katsuya Saito, Shunsuke Shibao, Hideaki Nagashima, Ryota Tamura, Yukina Morimoto, Hikaru Sasaki, Shinobu Noji, Yutaka Kawakami, Kazunari Yoshida, Masahiro Toda

**Affiliations:** 1Department of Neurosurgery, Hiratsuka City Hospital, Hiratsuka, Kanagawa 254-0019, Japan; fi020084@yahoo.co.jp; 2Department of Neurosurgery, Keio University School of Medicine, Shinjuku, Tokyo 160-8587, Japan; ueda.ryo@gmail.com (R.U.); 032m2068@gmail.com (H.N.); moltobello-r-610@hotmail.co.jp (R.T.); yukinaxnashiko@yahoo.co.jp (Y.M.); hsasaki@a5.keio.jp (H.S.); Kazrmky@keio.jp (K.Y.); 3Department of Neurosurgery, Kawasaki Municipal Hospital, Kawasaki, Kanagawa 210-0013, Japan; 4Department of Neurosurgery, Ashikaga Red Cross Hospital, Ashikaga, Tochigi 326-0843, Japan; csrbb415@yahoo.co.jp (K.S.); pochisuke616@mac.com (S.S.); 5Division of Cellular Signaling, Institute for Advanced Medical Research, Keio University School of Medicine, Shinjku, Tokyo 160-8587, Japan; snoji@keio.jp (S.N.); yutakawa@keio.jp (Y.K.)

**Keywords:** vaccine therapy, oncoantigen, tumor associate antigen, tumor angiogenesis, high-grade glioma

## Abstract

High-grade gliomas (HGGs) carry a dismal prognosis despite current treatments. We previously confirmed the safety and immunogenicity of a vaccine treatment targeting tumor angiogenesis with synthetic peptides, for vascular endothelial growth factor receptor (VEGFR) epitopes in recurrent HGG patients. In this study, we evaluated a novel vaccine therapy targeting not only tumor vasculature but also tumor cells, using multiple glioma oncoantigen (GOA)/glioma angiogenesis-associated antigen (GAAA) peptides in HLA-A2402+ recurrent/progressive HGG patients. The vaccine included peptide epitopes from four GOAs (LY6K, DEPDC1, KIF20A, and FOXM1) and two GAAAs (VEGFR1 and VEGFR2). Ten patients received subcutaneous vaccinations. The primary endpoint was the safety of the treatment. T-lymphocyte responses against GOA/GAAA epitopes and treatment response were evaluated secondarily. The treatment was well tolerated without any severe systemic adverse events. The vaccinations induced immunoreactivity to at least three vaccine-targeted GOA/GAAA in all six evaluable patients. The median overall survival time in all patients was 9.2 months. Five achieved progression-free status lasting at least six months. Two recurrent glioblastoma patients demonstrated stable disease. One patient with anaplastic oligoastrocytoma achieved complete response nine months after the vaccination. Taken together, this regimen was well tolerated and induced robust GOA/GAAA-specific T-lymphocyte responses in recurrent/progressive HGG patients.

## 1. Introduction

High-grade gliomas (HGGs) carry a dismal prognosis despite current treatments [1,2,3,4]. Options are particularly limited for patients with recurrent HGGs so new therapies are needed. Cancer vaccines are promising in this regard, designed to induce systemic immunity against antigens overexpressed by tumor cells and other components in the tumor microenvironment. Pilot clinical trials by us and others have exhibited the safety and potential efficacy of cytotoxic T lymphocyte (CTL) epitope peptide-based vaccinations for patients with HGGs [5,6,7,8,9,10,11].

Although cancer vaccines have been anticipated as a promising modality to treat cancer, recent reports indicated several mechanisms in tumor tissues that protect cancer cells from immune attacks [12]. For example, the limitation of the antitumor effects of CTLs was explained by inter- and intra-tumoral heterogeneity; a subset of tumor cells revealed downregulation, or loss of expression of human leukocyte antigen (HLA), or targeted antigen proteins [13,14]. To overcome the suppression of CTL antitumor effects, which occur due to tumor cell heterogeneity, we previously focused on a peptide vaccine targeting the tumor vasculature in the tumor microenvironment and demonstrated the safety and immunogenicity of vaccination with synthetic peptides for vascular endothelial growth factor receptor (VEGFR) epitopes in recurrent HGG patients [11].

Targeting of multiple glioma antigen epitopes also helps to address the issue of inter- and intra-tumoral heterogeneity of glioma cells. Furthermore, “oncoantigens” are ideal targets for a cancer vaccine [15,16,17,18,19,20,21] as they are essential for cell growth, and the probability of immune escape of cancer cells by reducing or lacking these proteins is expected to be low [22]. Therefore, this clinical trial was based on the use of HLA-A2402–restricted CTL epitopes derived from four glioma oncoantigens (GOAs) that we and others observed to be highly expressed in HGGs [23,24,25,26]: Lymphocyte antigen 6 family member K (LY6K), DEP domain containing 1 (DEPDC1), kinesin family member 20A (KIF20A), and forkhead box M1 (FOXM1)—in addition to two glioma angiogenesis-associated antigen (GAAAs): VEGFR1 and VEGFR2 [27,28].

To the best of our knowledge, this is the first study to evaluate a glioma vaccine therapy targeting tumor vasculature, as well as tumor cells with multiple glioma antigen epitope peptides derived from glioma cell-expressed oncoantigens and glioma angiogenesis factors. The primary objectives were to assess the tolerability of this regimen and its ability to induce GOA/GAAA epitope-specific immune responses.

## 2. Materials and Methods

The study protocol was approved by the institutional ethics committee (#20130294).

### 2.1. Vaccine Therapy Design

This study was a non-randomized, open label clinical trial with cocktail peptide vaccines for recurrent/progressive HGGs. The primary endpoint of this study was the safety of the peptide vaccine treatment. Secondary endpoints were the GOA/GAAA epitope–specific immune responses and the therapeutic outcome of patients treated with this vaccine.

### 2.2. Patient Eligibility

As we wished to focus on safety and immunoreactivity to the antigens in this vaccine treatment, we enrolled patients with recurrent/progressive HGG (World Health Organization (WHO) grade III/IV glioma) including, but not limited to, glioblastoma (grade IV glioma) from April 2014 to November 2016 at Keio University Hospital (Tokyo, Japan)—resulting in a somewhat heterogenous patient cohort.

Inclusion criteria were as follows: (1) histological diagnosis of supratentorial HGG (World Health Organization (WHO) grade III or IV according to the 2007 WHO criteria) without multiple lesions or leptomeningeal dissemination; (2) patients were informed about their diagnosis; (3) HLA-A*2402-positive status; (4) age between 16 and 79 years; (5) Eastern Cooperative Oncology performance status 0–2; (6) completion of standard treatment (surgical therapy + radiation therapy + temozolomide); (7) four-week interval from last chemotherapy or radiotherapy; (8) adequate bone-marrow, cardiac, pulmonary, and hepatic and renal functions including neutrophil ≥1000/μL, platelet count ≥50,000/μL, hemoglobin ≥8 g/dL, plasma aspartate aminotransferase and alanine aminotransferase levels ≤4 times the normal limit, plasma bilirubin levels ≤1.5 times the normal limit, plasma albumin levels ≥2.5 g/dL, and plasma creatinine levels ≤2.0 mg/dL; (9) life expectancy >3 months; (10) signature confirming informed consent. Exclusion criteria were as follows: (1) uncontrollable infection; (2) the presence of another serious disease such as uncontrolled diabetes, hepatic disorder, cardiac disease, hemorrhage/bleeding; (3) total parenteral nutrition; (4) multiple cancers; (5) myelodysplastic syndrome (MDS), MDS/myeloproliferative disease (MPD) and MPD; (6) allogenic hematopoietic stem cell transplantation; (7) severe immunological disorders (autoimmune disease, immunosuppression); (8) anaphylaxis to synthetic peptides; (9) concurrent treatment with steroids or immunosuppressive agents; (10) pregnant or breast-feeding women; (11) severe mental disorder; (12) unhealed wound; (13) decision of unsuitability by the principal investigator or the physician in charge.

### 2.3. Peptides

The peptide vaccine included HLA-A2402-restricted epitopes for four GOAs (LY6K, DEPDC1, KIF20A, and FOXM1) and two GAAAs (VEGFR1 and VEGFR2). These peptide epitopes have been previously identified and evaluated for safety and potent immunogenicity in various cancer patients: a VEGFR1-derived peptide (VEGFR1-1084; SYGVLLWEI) [29], a VEGFR2-derived peptide (VEGFR2-169; RFVPDGNRI) [30], a LY6K-derived peptide (LY6K-177; RYCNLEGPPI) [31], a DEPDC1-derived peptide (DEPDC1-294; EYYELFVNI) [16], a KIF20A-derived peptide (KIF20A-66; KVYLRVRPLL) [32], and a FOXM1-derived peptide (FOXM1-262; IYTWIEDHF) [33]. All GMP-grade peptides were synthesized by the American Peptide Company (Sunnyvale, CA, USA) according to a standard solid-phase synthesis method and purified by reversed-phase high-performance liquid chromatography (HPLC). The purity (>90%) and identity of the peptides were determined by analytical HPLC and mass spectrometry analysis, respectively.

### 2.4. Vaccine Preparation and Treatment Protocol

One milligram of each peptide was emulsified in incomplete Freund’s adjuvant (Montanide ISA-51VG; SEPPIC, Paris, France) and administered subcutaneously close to an axillary or inguinal lymph node, eight times weekly. Patients demonstrating no clinical or radiological progression without adverse events had the option of continuing to receive vaccinations at 2-week intervals, for up to 8 months after the initial vaccination.

### 2.5. Radiologic Response Monitoring and Other Clinical Endpoints

Tumor size was assessed at weeks 9, 17, 25, and 33, then every 3 months thereafter using magnetic resonance imaging (MRI) with contrast enhancement. Response was evaluated by the Response Evaluation Criteria in Solid Tumors [34] and Immunotherapy Response Assessment in Neuro-Oncology [35] by gadolinium-enhanced T1 weighted images on the basis of the appearance of the pretreatment MRI. Overall survival (OS) was defined by the interval from initial vaccination to date of death. MRI was used to evaluate tumor progression over time.

### 2.6. Toxicity Assessment

Toxicity was assessed based on the common terminology criteria for adverse effects version 4.0. Toxicity was defined as toxicity of grade 4 or greater.

### 2.7. CTL Responses to Peptide Stimulation

To evaluate the specific CD8+ T-cell response, an enzyme-linked immunosorbent spot (ELISPOT) assay was performed in six cases using a procedure reported in a prior study [11].

### 2.8. Statistical Analysis

Statistical analyses were performed using SPSS 24.0 software (IBM, Chicago, IL, USA). OS curves were estimated using Kaplan–Meier methodology. Statistical analyses were performed with the log-rank test and differences were considered statistically significant at *p* < 0.05.

## 3. Results

### 3.1. Demographics and Clinical Characteristics

A total of 10 patients—who were found to be HLA-A2402 positive by DNA typing of HLA genomic variations—were enrolled in this study. Three patients were initially treated in other hospitals. Mean age was 44 years old (range, 17–72). Mean follow-up was 16.2 months (range, 3.6–38.1). Seven of the 10 patients were diagnosed with glioblastoma. Table 1 shows the characteristics of the 10 enrolled patients.

### 3.2. Toxicity

No severe adverse events associated with the vaccine were observed. During the vaccination therapy, skin flare (grade 1) was shown in one patient and induration (grade 1) was shown in five patients at the injection site. Wound infection (grade 2) and herpes zoster (grade 2) were each found in a single patient during the observation period and were considered to be unrelated to the vaccination.

### 3.3. CTL Response

CTL responses were analyzed in six evaluable patients, as shown in Table 2. All six patients showed specific CTL responses to at least three vaccine-targeted GOA/GAAA epitopes.

### 3.4. Clinical Outcomes

Although the primary goal of this study was to provide an analysis of safety and immunoreactivity, preliminary outcome data were obtained (Table 3 and Figure 1). Patients received a mean of 14.2 (range, 8–26) peptide vaccinations. One patient achieved partial response (PR), two patients demonstrated stable disease, and six patients revealed progressive disease 6 months after the first vaccination (Table 3). Patient 7 was removed from the study due to rapid tumor progression. Patients 3, 6, and 10 remain progression-free at 18, 38, and 11 months, respectively, after the first vaccination. Among these patients, Patient 6 achieved compete response (CR) 9 months after the first vaccination. These results indicate the preliminary efficacy of this treatment.

The Kaplan–Meier curves for overall survival in all 10 patients and seven glioblastoma (GB) patients are shown in Figure 1a,b, respectively. The median overall survival time (mOS) in all patients and GB patients was 9.2 months and 9.1 months, respectively. One-year OS was 44.4% for all patients and 33.3% for GB patients, respectively.

Five patients were treated with bevacizumab before registration. In this group, 1-year OS was 0% and mOS was 8.6 months. Otherwise, in GB patients who had not received bevacizumab before registration, mOS was 23.6 months. Our findings suggest that the GB patients who did not receive bevacizumab had a longer survival period than those treated with bevacizumab following a combination of chemotherapy and/or radiotherapy, but no significant differences in OS were observed—likely due to the small sample numbers.

### 3.5. A Case of CR following Peptide Vaccination

Patient 6 was a 33-year-old female diagnosed with diffuse astrocytoma (grade 2) four years prior. Her tumor was enlarged and removed twice, followed by treatment with TMZ and radiation therapy for the preceding 12 months. The pathological diagnosis was anaplastic oligoastrocytoma (grade 3, MGMT unmethylated, IDH mutant and no 1p19q codeletion). However, her tumor recurred and could not be removed as it was located in a functional area (Figure 2a). She was thus enrolled in our study. Her tumor decreased in size three months after vaccine initiation and disappeared nine months after enrollment (Figure 2b,c). Thirty-eight months after the initiation of peptide vaccination, the patient remains free of tumor recurrence.

## 4. Discussion

This is the first clinical evaluation of peptide-based vaccine therapy, targeting glioma cells as well as glioma neovascular endothelial cells, using multiple GOA/GAAA-derived epitopes for recurrent/progressive HGG. Our findings demonstrate tolerability and immunoreactivity to GOAs/GAAAs, as well as the preliminary efficacy of this treatment.

The population was very small and not homogeneous in this study. However, this was a pilot study to assess safety and immunoreactivity to the antigens, which allowed us to assess the tolerability and immune response regardless of patient characteristics.

The peptide epitopes included in this vaccine treatment were derived from six proteins known as GOAs or GAAAs [23,24,25,26,27,28]. ELISPOT data demonstrated that all evaluable vaccinated patients mounted an immune response against at least three of the target antigens, supporting the use of such epitopes in glioma vaccine regimens. ELISPOT data also showed that CTLs specific for three oncoantigens, DEPDC1, FOXM1, and LY6K were frequently observed in peripheral blood mononuclear cells from the vaccinated patients—indicating that these oncoantigens are highly immunogenic in advanced HGG patients. To evaluate if the induced CTLs contributed to reduction of tumor cells or tumor vascular endothelial cells in the microenvironment, further immunohistochemical analyses of tumor tissues obtained from vaccinated patients or blood flow analyses that can detect hypoperfusion peri-/intra-tumorally are warranted.

Although this was a pilot study focusing on safety and immunoreactivity to the antigens, we also evaluated treatment response in the vaccinated patients. In this study, the mOS in all patients and GB patients was 9.2 months and 9.1 months, respectively. Our median survival results are comparable to, but do not exceed those reported in the literature by previous clinical studies of glioma vaccines [9,10] and various combination regimens of chemotherapy and/or radiotherapy for recurrent GB patients [36,37,38,39]. This may be reflected in immune tolerance or a hostile immune status mediated by regulatory T-cell populations or tumor-secreted immunosuppressive factors in immunocompromised patients with recurrent HGG. Previous studies have demonstrated that anti-VEGF agents, such as bevacizumab, inhibit proliferation of immunosuppressive cells, such as regulatory T-cells and myeloid derived suppressor cells [40,41,42]—suggesting that VEGF-VEGFR pathway blockade could restore and improve antitumor immune responses. Nevertheless, the HGG patients who did not receive bevacizumab had a longer survival period than the patients treated with bevacizumab following a combination of chemotherapy and/or radiotherapy, although the sample size was relatively small (Figure 3). These results suggest that such approaches may be most effective if applied early in treatment, particularly in patients likely to have a robust immunity, such as the patients who have not yet received any chemotherapy or radiation therapy. In fact, cancer vaccines often need more time to elicit beneficial immune responses that demonstrate biological activity, which is shown by the occurrence of delayed vaccine effects.

One HGG patient (Patient 6) experienced objective clinical tumor regression (response rate of this vaccine treatment was 10%). Furthermore, it is noteworthy that this patient exhibited PR at week six and CR at week nine. CTLs specific for all six antigens were strongly induced in the patients, suggesting that this CTL response might contribute to the observable effect.

As biological features of HGGs in children are different from those that arise in adults [43,44], it is necessary to discuss the cases of children specifically. Our heterogenous patient cohort included a 17 year-old patient, whose OS was 8.9 months after enrollment. This vaccine therapy could not extend OS significantly, but could safely induce CTLs specific for three oncoantigens in this patient, suggesting that this vaccine therapy theoretically has the potential to exert an antitumor effect for pediatric HGGs expressing target antigens. Therefore, immunoreactivity to antigens and clinical efficacy of this regimen for children with HGGs will be assessed in a future study.

In summary, we performed a pilot study for HLA-A2402+ patients with recurrent/progressive HGGs to assess the safety, feasibility, and immunoreactivity of the peptide-based vaccine targeting GOAs and GAAAs. The safety and immunogenicity of this vaccine therapy was verified. The data suggest that this vaccine treatment may show preliminary evidence of clinical responses. However, a future study of this vaccine in combination with standard treatment for newly-diagnosed HGGs as well as immune-checkpoint blockade therapies, is required to improve the efficacy of glioma vaccine therapy.

## Figures and Tables

**Figure 1 jcm-08-00263-f001:**
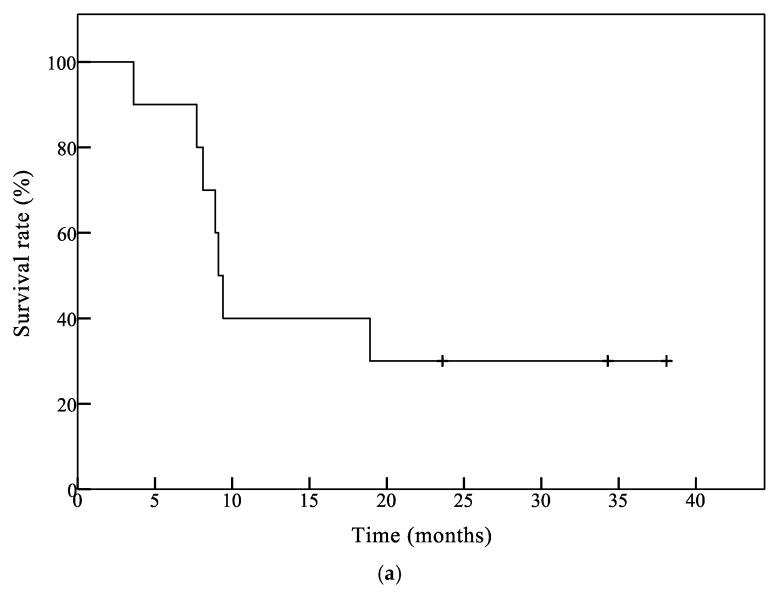
Survival analysis of patients by the Kaplan–Meier method. (**a**) Overall survival (OS) curve of all patients (*n* = 10). The median OS time (mOS) of all patients was 9.2 months and 1-year OS was 44.4%; (**b**) OS curve of glioblastoma (GB) patients (*n* = 7). The mOS was 9.1 months and 1-year OS was 33.3% in GB patients.

**Figure 2 jcm-08-00263-f002:**
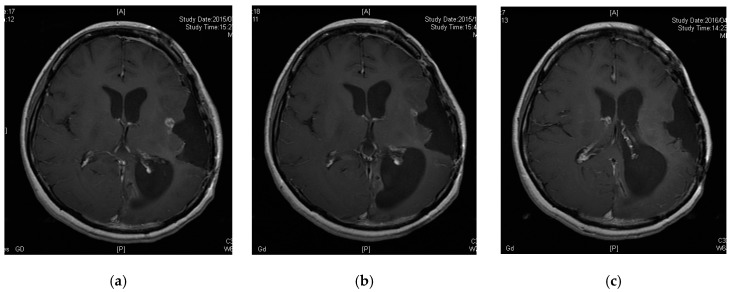
Contrast-enhanced magnetic resonance images of Patient 6. (**a**) Tumor had recurred in a functional area; (**b**) tumor was decreased 3 months after enrollment; (**c**) tumor disappeared 9 months after enrollment.

**Figure 3 jcm-08-00263-f003:**
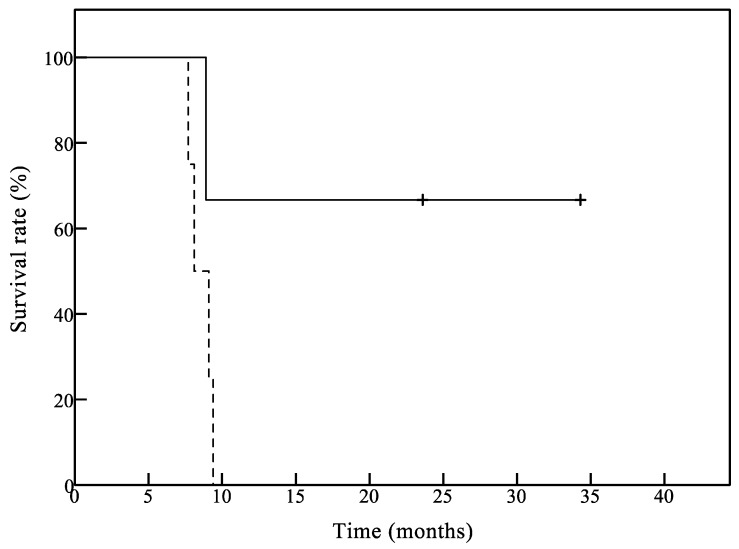
Overall survival of glioblastoma patients with or without bevacizumab. The median overall survival time (mOS) was 23.6 months in three patients that did not receive bevacizumab before enrollment (solid line). The mOS was 8.6 months in four patients treated with bevacizumab before enrollment (dotted line).

**Table 1 jcm-08-00263-t001:** Patient characteristics

Case No.	Age (Years)	Sex	Diagnosis	Tumor Size (mm)	Operation	Radiotherapy	Chemotherapy	IDH1 Mutation	1p/19q Codeletion	MGMT Methylation
1	17	M	GB	32.5 × 29.5 × 32.5	2	60 Gy	TMZ, ICE	WT	(−)	(−)
2	38	M	HGG	20.6 × 11.7 × 16.6	0	60 Gy + SRT30 Gy	TMZ	NT	NT	NT
3	38	M	GB with oligo	No enhanced lesion *	2	60 Gy	TMZ	WT	NT	(−)
4	66	F	GB	18.0 × 11.5 × 20.0	1	60 Gy	TMZ, BEV	NT	NT	NT
5	46	F	sGB	48.0 × 25.0 × 48.8	5	GK, 60 Gy	TMZ, IFNb, BEV	NT	NT	NT
6	33	F	AOA	12.0 × 8.5 × 18.0	4	60 Gy	TMZ	R132H	(−)	(−)
7	72	M	OA rec	33.0 × 20.5 × 23.7	1	60 Gy	TMZ, BEV	WT	(−)	(±)
8	36	F	sGB	16.0 × 13.0 × 18.6	2	60 Gy	TMZ, BEV	R132H	(−)	(−)
9	27	F	sGB	30.2 × 24.1 × 28.6	1	SRT	TMZ, BEV	R132H	(−)	(+)
10	67	F	GB	23.3 × 12.1 × 17.5	2	60 Gy	TMZ	WT	(−)	(±)

AOA, anaplastic oligoastrocytoma; BEV, bevacizumab; F, female; GB, glioblastoma; GB with oligo, glioblastoma with oligodendroglial component; GK, gamma knife; HGG, high grade glioma; ICE, ifosfamide, carboplatin, and etoposide; IDH, Isocitrate dehydrogenase; IFNb, interferon beta; M, male; MGMT, O-6-methylguanine-DNA-methyl-transferase; NT, not tested; OA rec, recurrent oligoastrocytoma; sGB, secondary glioblastoma; SRT, stereotactic radiotherapy; TMZ, temozolomide; WT, wild type. * This patient was enrolled after complete recurrent tumor removal.

**Table 2 jcm-08-00263-t002:** Cytotoxic T lymphocyte (CTL) responses to target antigens.

Case No.	Vaccination	LY6K	FOXM1	DEPDC1	KIF20A	VEGFR1	VEGFR2	Positive Control
1	before	−	+	+	+	+	−	+++
	2 weeks after	+++	+++	+++	+	+	−	+++
2	before	−	−	+	+	−	−	+++
	2 weeks after	+++	+++	+++	+	−	−	+++
3	before	−	−	NT	NT	−	NT	+++
	2 weeks after	+	+++	+	−	+	NT	+++
4	before	−	+	+	−	−	+	+++
	2 weeks after	+++	+++	+++	+	−	+	+++
5	before	−	+	−	−	−	−	+++
	2 weeks after	+++	+++	+++	+	+	+++	+++
6	before	−	+	−	−	−	−	+++
	2 weeks after	+++	+++	+++	+	+	+++	+++

NT, not tested.

**Table 3 jcm-08-00263-t003:** Clinical results of 10 enrolled patients.

Case No.	Frequency of Vaccination	Period of Vaccination (mo)	Evaluation after 3 Months	Evaluation after 6 Months	PFS (mo)	OS (mo)
1	18	6.2	PD	PD	6.3	8.9
2	11	6.7	PD	PD	6.8	18.9
3	26	21.0	SD	SD	18.2	34.3
4	12	4.8	PD	PD	4.9	9.1
5	8	1.6	PD	PD	1.7	8.1
6	20	37.5	PR	PR *	38.1	38.1
7	8	1.6	PD	Dead	1.9	3.6
8	11	4.6	SD	PD	4.7	7.7
9	10	2.1	PD	PD	2.9	9.4
10	18	10.8	SD	SD	11.0	23.6

Mo, months; OS, overall survival; PD, progressive disease; PFS, progression-free survival; PR, partial response; SD, stable disease. * Complete response was achieved after 9 months.

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
