# Peer review of "A Pilot Study of Vaccine Therapy with Multiple Glioma Oncoantigen/Glioma Angiogenesis-Associated Antigen Peptides for Patients with Recurrent/Progressive High-Grade Glioma"

_jcm, 2019, doi:10.3390/jcm8020263_

Round 1
Reviewer 1 Report
I red with great interest the pilot study of vaccine therapy against high grade glioma onco-antigens and angiogenesis-associated antigens. The study received proper ethical comitee aproval and it has been well defined and explained. Unfortunately the population is very small (only 10 patients) and it is not homogeneous as regards age and disease but it is justifiable as this is a unique pilot study. I would suggest to spend some words to explain this lack of homogeneity. Moreover, it would be interesting if authors compare the overall survival of their population which has been treated by vaccine therapy to another comparable population of high grade glioma patients who underwent only surgery and adjuvant chemotherapy and radiotherapy.
Author Response
Response to Reviewer 1
We truly appreciate the reviewers’ kind review and comments on our manuscript entitled “A pilot study of vaccine therapy with multiple glioma oncoantigen/glioma angiogenesis-associated antigen peptides for patients with recurrent/progressive high-grade glioma” (Manuscript number: jcm-434689) following our submission to the Journal of Clinical Medicine. We have addressed each comment, point by point below and revised our manuscript accordingly. As we had omitted Figure 2 from the original manuscript, despite attaching the file containing this figure, we have now included Figure 2 in the manuscript and renumbered the remaining figures accordingly.
Point 1: Unfortunately, the population is very small (only 10 patients) and it is not homogeneous as regards age and disease but it is justifiable as this is a unique pilot study. I would suggest to spend some words to explain this lack of homogeneity.
[Response]: As we wished to focus on safety and immunoreactivity to the antigens of this vaccine treatment, we enrolled various patients with recurrent/progressive high-grade glioma (WHO grade III/IV glioma) including, but not limited to, glioblastoma (grade IV glioma). As a result, the registered patient population became heterogeneous. We have now included this description in the “Materials and Methods” section.
Point 2: Moreover, it would be interesting if authors compare the overall survival of their population which has been treated by vaccine therapy to another comparable population of high grade glioma patients who underwent only surgery and adjuvant chemotherapy and radiotherapy.
[Response]: In agreement with the reviewer’s point, we have cited further previous publications by others that employed combination regimens of chemotherapy and/or radiotherapy for recurrent high-grade glioma patients. We have also expanded our discussion on this matter in the Discussion section (page 7, line220-225).
We are very grateful to the reviewers for their helpful comments, and we hope that the revised manuscript meets the high standard for publication in the Journal of Clinical Medicine.
Reviewer 2 Report
The study in the manuscript by Kikuchi et al. was focused on evaluating tolerance and immunoreactivity of a vaccine therapy by using multiple glioma oncoantigen and angiogenesis-associated antigen peptides in a small size of patients with glioma. The authors demonstrated lymphocyte response against the peptide epitopes from four GOAs (LY6K, DEPDC1, KIF20A, and 25 FOXM1) and two GAAAs (VEGFR1 and VEGFR2) in three patient. Although this pilot study was performed in very small size cohort of patient, the multiple antigen peptide vaccine could be a safe and effective therapy for the treatment of glioma. Below are several comments/criticisms of this work:
1) The authors should show immunohistochemistry staining of these antigens in these patient biospecimens, which could confirm whether patient with high level of the antigens expression have a better response than those with low level of these antigens.
2) Regarding very limited number of patients, multiple factors. e.g. age, sex, tumor grade, treatment, can affect patient outcome. The authors should clearly and carefully describe each patient’s outcome after receiving these peptide vaccinations. For example, patient 6 showed good response in the study, which resulted from IDH1 mutation, or longest period vaccination vs other patients (37.5 months).
3) The authors mentioned patients without bevacizumab treatment before vaccination showed a longer survival period than those with bevacizumab treatment. This is an interesting observation. does this because of beva treatment reduce VEGFR1/2 expression or normalized blood vessels by beva treat limit T lymphocyte recruited to tumor sites, which led to resistance to peptide vaccination? it is better to provide further evidence or discuss in detailed.
Author Response
Response to Reviewer 2
We truly appreciate the reviewers’ kind review and comments on our manuscript entitled “A pilot study of vaccine therapy with multiple glioma oncoantigen/glioma angiogenesis-associated antigen peptides for patients with recurrent/progressive high-grade glioma” (Manuscript number: jcm-434689) following our submission to the Journal of Clinical Medicine. We have addressed each comment, point by point below and revised our manuscript accordingly. As we had omitted Figure 2 from the original manuscript, despite attaching the file containing this figure, we have now included Figure 2 in the manuscript and renumbered the remaining figures accordingly.
Although this pilot study was performed in very small size cohort of patient, the multiple antigen peptide vaccine could be a safe and effective therapy for the treatment of glioma. Below are several comments/criticisms of this work:
Point 1: The authors should show immunohistochemistry staining of these antigens in these patient biospecimens, which could confirm whether patient with high level of the antigens expression have a better response than those with low level of these antigens.
[Response]: We agree with the reviewer’s suggestion that investigation of the association between the level of antigen expression and immunoreactivity to the antigen by methods such as immunohistochemistry would be highly meaningful. While we and others have already observed the selected glioma oncoantigens to be highly expressed in high-grade gliomas, we were intrigued by the antigen protein expression in tumor tissues of the patients treated with our vaccine therapy. However, in most of our enrolled patients, we could only obtain blood samples and not tumor tissue. Therefore, it was not possible to analyse expression of these antigens in tumor tissues, or correlate antigen expression with immune response. However, we plan to obtain paired samples of tumor tissues and sera to evaluate this relationship in additional cases in future studies.
Point 2: Regarding very limited number of patients, multiple factors. e.g. age, sex, tumor grade, treatment, can affect patient outcome. The authors should clearly and carefully describe each patient’s outcome after receiving these peptide vaccinations. For example, patient 6 showed good response in the study, which resulted from IDH1 mutation, or longest period vaccination vs other patients (37.5 months).
[Response]: We agree with the reviewer’s point. We have provided the clinical course of Patient 6 in addition to Table 3, which shows the patient’s characteristics. Relevant sentences have been added to the “Materials and Methods” and “Results” sections (page2, line76-78; page 7, line190-197).
Point 3: The authors mentioned patients without bevacizumab treatment before vaccination showed a longer survival period than those with bevacizumab treatment. This is an interesting observation. does this because of beva treatment reduce VEGFR1/2 expression or normalized blood vessels by beva treat limit T lymphocyte recruited to tumor sites, which led to resistance to peptide vaccination? it is better to provide further evidence or discuss in detailed.
[Response]: Previous studies have demonstrated that anti-VEGF agents inhibit proliferation of immunosuppressive cells, such as regulatory T-cells and myeloid derived suppressor cells, suggesting that VEGF-VEGFR pathway blockade could restore and improve antitumor immune responses. Nevertheless, the HGG patients who did not receive bevacizumab had a longer survival period than the patients treated with bevacizumab following combination of chemotherapy and/or radiotherapy. We think that this may be reflected in immune tolerance in immunocompromised recurrent HGG patients with advanced disease status and already resistant to conventional treatments before enrollment. In agreement with the reviewer’s point, we have further cited previous publications by others and expanded our discussion of this matter in the Discussion section (page 8, line 227-230, 236-7).